# Temperature-Resistant Intrinsic High Dielectric Constant Polyimides: More Flexibility of the Dipoles, Larger Permittivity of the Materials

**DOI:** 10.3390/molecules27196337

**Published:** 2022-09-26

**Authors:** Weiwen Zheng, Zuhao Li, Kaijin Chen, Siwei Liu, Zhenguo Chi, Jiarui Xu, Yi Zhang

**Affiliations:** PCFM Lab, GD HPPC Lab, Guangdong Engineering Technology Research Centre for High-Performance Organic and Polymer Photoelectric Functional Films, State Key Laboratory of Optoelectronic Materials and Technologies, School of Chemistry, Sun Yat-sen University, Guangzhou 510275, China

**Keywords:** polyimide, high dielectric constant, excellent thermal stability, high discharged energy density, flexible of the dipoles

## Abstract

High dielectric constant polymers have been widely studied and concerned in modern industry, and the induction of polar groups has been confirmed to be effective for high permittivity. However, the way of connection of polar groups with the polymer backbone and the mechanism of their effect on the dielectric properties are unclear and rarely reported. In this study, three polyimides (C0-SPI, C1-SPI, and C2-SPI) with the same rigid backbone and different linking groups to the dipoles were designed and synthesized. With their rigid structure, all of the polyimides show excellent thermal stability. With the increase in the flexibility of linking groups, the dielectric constant of C0-SPI, C1-SPI, and C2-SPI enhanced in turn, showing values of 5.6, 6.0, and 6.5 at 100 Hz, respectively. Further studies have shown that the flexibility of polar groups affected the dipole polarization, which was positively related to the dielectric constant. Based on their high permittivity and high temperature resistance, the polyimides exhibited outstanding energy storage capacity even at 200 °C. This discovery reveals the behavior of the dipoles in polymers, providing an effective strategy for the design of high dielectric constant materials.

## 1. Introduction

With the continuous development of electronic devices and the electronics industry, dielectric capacitors have shown great potential and impact in photovoltaic devices [1,2,3], new energy vehicles [4,5,6], inverters [7,8,9], and other industries. Compared to inorganic dielectrics [10,11,12] and polymer nanocomposites [13,14,15,16,17,18], intrinsic polymer dielectrics have gradually become the research hotspot because of their good structural designability, flexibility, solution processing, low processing temperature, and high breakdown field strength. One of the most important traditional polymer dielectrics used in film capacitors is biaxially oriented polypropylene (BOPP), which has a high breakdown strength and a low dielectric loss. However, the low dielectric constant of BOPP leads to its low energy density. In addition, its low operating temperature also limits its applications in fields of extremely high temperature [19,20,21,22]. Therefore, it is very necessary to develop new dielectric polymers for film capacitors with a high dielectric constant, low dielectric loss, and high temperature resistance.

To enhance the permittivity of intrinsic polymer materials, many researchers have adopted different strategies to design molecular structures, including self-organized nanostructured polymers [23,24,25,26], ionic polymers [27,28], metal coordination polymers [29,30], and dipolar polymers [31,32,33,34,35]. Among them, introducing high dipole moment units such as nitro groups [36], nitrile groups [37,38,39,40], and sulfone groups [34,41,42,43,44] to the side chain is one of the most effective methods. This is because the dipole moment is the nature of polymers, which can be designed and adjusted easily while maintaining their properties. As is known, the molecular structure of materials determines their properties. Thus, the relationship between the molecular structure and the dielectric properties is important for the design of new dielectrics. However, although various polar groups have been introduced into different polymers, the influence of the linking groups between the polar groups and the polymer backbone on the dielectric properties of the materials is still unclear and rarely reported.

Among various polymer materials, polyimides (PI) have been a topic of wide concern due to their excellent mechanical properties, good film forming, and especially their excellent thermal stability. Therefore, many researchers have developed different kinds of intrinsic high dielectric constant (high-*k*) polyimides [40,45,46,47,48]. Previously, by introducing flexible polar side groups to the rigid backbones, we designed and synthesized a novel intrinsic high-*k* polyimide [47]. Compared with the counterpart without the polar side group, the movable polar side groups provide effective dipole polarization following the electric field, resulting in both high and stable permittivity (5.2~6.0 from 1 MHz to 100 Hz) and excellent thermal stability (*T*_d5%_ > 450 °C and *T*_g_ > 300 °C).

In this study, the influence of the flexibility of the polar side groups on the dielectric properties is further discussed. Three polyimides (C0-SPI, C1-SPI, and C2-SPI) with the same rigid backbones and different linking groups were successfully designed and synthesized. All of the polyimides exhibited excellent thermal stability and high dielectric constant. With the increase in the flexibility of the side chains in C0-SPI, C1-SPI, and C2-SPI, the dielectric constant enhanced in turn. Experimental results and theoretical simulations systematically revealed that the improvement of the side chain flexibility enhances the orientation ability of the dipoles, thus increasing the dipole polarizability. When applied for energy storage, all of the polyimides delivered high discharged energy densities even at 200 °C. 

## 2. Results and Discussion

### 2.1. Characterization of Polyimides

The chemical structures of C0-SPI, C1-SPI, and C2-SPI are shown in Figure 1a. All of the polyimides contain the same rigid backbones, but their side groups show different alkyl chain lengths. Among them, the methyl sulfone group is connected to the main chain directly in C0-SPI, while by one methylene group in C1-SPI and two methylene groups in C2-SPI. The possible stable conformations of their corresponding repetitive units, C0 unit, C1 unit, and C2 unit, are shown in Figure 1b via a conformation search. It can be found that longer alkyl chains will lead to more conformations, indicating more flexible side chains. All the polyimides were prepared by monomer synthesis and mild polycondensation. Because they were insoluble in THF, DMF, and DMAc, their molecular weights were indirectly reflected by measuring the molecular weights of corresponding precursors PAA. The average molecular weights (*M*_n_) of the three polymers are 305 kDa (PDI = 1.21), 329 kDa (PDI = 1.33), and 312 kDa (PDI = 1.24), respectively. The high molecular weights of the precursors ensure the quality of the polyimide films, and all of them can be prepared as large-area self-standing films.

Fourier infrared spectroscopy (FT-IR) was performed to characterize the chemical structure of the obtained polyimide films, as shown in Appendix A. Infrared absorption peaks of the three polyimides at 1776–1775 cm^−1^, 1717–1716 cm^−1^, and 1367–1358 cm^−1^ indicate the imide ring structures and the formation of polyimides. In addition, the absorption peaks at 1310–1300 cm^−1^ and 1157 cm^−1^ correspond to the antisymmetric and symmetric stretching vibrations of -SO_2_-, confirming the successful introduction of the sulfone groups into the three materials.

The morphologies of PI films were analyzed via X-ray diffraction (XRD) (Appendix A). The three polyimides all exhibit diffuse broad peaks, indicating that they are amorphous structures. The maximum 2*θ* of C0-SPI and C1-SPI is around 19°, while C2-SPI is around 16°. The corresponding *d*-spacing according to Bragg’s law is 4.67 Å for C0-SPI and C1-SPI, and 5.54 Å for C2-SPI. C2-SPI has the largest full-width half-maximum due to the longest side group being in C2-SPI, which causes a looser chain arrangement and larger chain spacing, making it more prone to disorder.

The mechanical properties of the three polyimide films were measured via a tensile test. (Appendix A). Except for C0-SPI, the tensile strength of C1-SPI and C2-SPI is above 100 MPa, and the tensile modulus is above 2 GPa. The good mechanical properties can fulfill the requirements of practical processing and applications. It can be observed that as the length of the side chains increases, the tensile strength and elongation at break become larger. This might be due to the enhancement of the flexibility and entanglement of the polymer chains.

The thermal properties of the PIs were investigated through thermogravimetric analysis (TGA), differential scanning calorimetry (DSC), and thermal mechanical analysis (TMA). All of the three materials exhibit excellent thermal stability, and the 5 wt% decomposition temperatures (*T*_d5%_) are all above 400 °C, which are 495 °C, 465 °C, and 448 °C, respectively (Figure 2a). Among them, C1-SPI and C2-SPI show obvious weight loss steps at 426–516 °C and 406–506 °C, and the corresponding weight loss ratio is about 14%, which corresponds to the decomposition of the side groups. C0-SPI has a higher decomposition temperature without an obvious decomposition step, which indicates that the structure with the sulfone group directly connected to the benzene ring has a higher thermal stability. In the DSC curves of the three polyimides (Figure 2b), no obvious glassy transition or melting peak show in the range of 50–300 °C, indicating that the glass transition temperature (*T*_g_) of the three polyimides is above 300 °C with good heat resistance. TMA results further investigate that the thermal expansion of the materials is basically linear at 50–350 °C, indicating that the *T*_g_ of the three materials is even above 350 °C (Figure 2c). Due to their rigid backbones, the three materials show good dimensional stability, and the coefficient of thermal expansion (CTE) is 44.4 μm∙m^−1^∙K^−1^, 40.7 μm∙m^−1^∙K^−1^, and 51.3 μm∙m^−1^∙K^−1^ at 100–300 °C, respectively.

### 2.2. Dielectric Properties of Polyimides

The dielectric properties of C0-SPI, C1-SPI, and C2-SPI were investigated through a parallel plate capacitor method via a precision impedance analyzer. A total of 50 nm gold electrodes were sputtered on both sides of the films to furnish a sandwich structure of Au/PI/Au devices. The results indicate that C0-SPI, C1-SPI, and C2-SPI with sulfone groups all show high permittivity at room temperature (Figure 3a,b). In the measurement range of 10^2^–10^6^ Hz, the dielectric constant of C0-SPI whose methyl sulfone group is directly connected to the benzene ring is the lowest, which is 5.6–5.2. The dielectric permittivity of C1-SPI is 6.0–5.3, whose methyl sulfone group and the benzene ring are connected by one methylene group. C2-SPI, with the methyl sulfoxide group connected to a benzene ring through two methylene groups, has the highest permittivity of 6.5–6.0. With the enhancement of the flexibility of side groups, the dielectric constant of the materials increases correspondingly. However, the increase in polarization also leads to an increase in dielectric loss slightly. At 100 Hz, the dielectric loss of the three polyimides is 0.009, 0.009, and 0.011, respectively. Because it is more difficult for the orientation polarization to keep up with the external electric field, the dielectric loss increases more obviously at 1 MHz, standing at 0.048, 0.057, and 0.071, respectively, for C0-SPI, C1-SPI, and C2-SPI.

The dielectric stability of C0-SPI, C1-SPI, and C2-SPI at different temperatures was further studied, as shown in Figure 4. In the range of −100–200 °C, the three polyimides all exhibit high permittivity. The permittivity of C0-SPI, C1-SPI, and C2-SPI at 100 Hz is 4.5–5.6, 4.6–6.3, and 5.1–6.5, respectively. Thanks to the excellent thermal stability of polyimide, the three dielectric materials can still work normally even at 200 °C, with dielectric constants of 4.7, 5.0, and 5.9, respectively. The dielectric loss at 10^2^–10^4^ Hz is 0.017–0.013, 0.025–0.011, and 0.080–0.014, respectively. Therefore, the three materials can be applied in fields of extremely high temperature, among which C0-SPI and C1-SPI exhibit lower losses at high temperatures. Dielectric loss–temperature curves indicate that all of the three polyimides show obvious secondary relaxation peaks at low temperatures, corresponding to the rotation of the side sulfone groups. The secondary relaxation behaviors can be well described by the Eyring equation [49,50], as shown in Appendix A. According to the fitting results, the activation energy of dipoles in C0-SPI, C1-SPI, and C2-SPI is 29.02 kJ∙mol^−1^, 39.53 kJ∙mol^−1^, and 36.62 kJ∙mol^−1^, respectively, while the relaxation time at room temperature (298.15 K) is 1.19 × 10^−7^ s, 3.68 × 10^−7^ s, and 2.23 × 10^−7^ s, respectively. The low activation energy and short relaxation time for the polar side groups in the polyimides endow them with high permittivity and low dielectric loss.

Theoretical simulations were performed to further study the molecular microstructure of the polyimides [51,52,53,54,55,56]. The Clausius–Mossotti equation is a classical model for describing a dielectric [57,58], revealing that the permittivity of a polymer is positively correlated with the number density and the average polarizability of the repetitive unit. Because of the similar chemical structures, C0-SPI, C1-SPI, and C2-SPI showed similar unit densities with 1.42 nm^−3^, 1.33 nm^−3^, and 1.36 nm^−3^, respectively. In the intrinsic polymer, the polarizability can be divided into an optical frequency part and a dipole part. For the repetitive units of C0-SPI, C1-SPI, and C2-SPI, they also exhibited similar optical frequency polarizability at 2.35 × 10^−38^ C∙m^2^∙V^−1^, 2.42 × 10^−38^ C∙m^2^∙V^−1^, and 2.48 × 10^−38^ C∙m^2^∙V^−1^, respectively. Furthermore, by molecular dynamic simulation, the average dipole polarizability of the repetitive units in C0-SPI, C1-SPI, and C2-SPI is 3.2 × 10^−39^ C∙m^2^∙V^−1^, 4.3 × 10^−39^ C∙m^2^∙V^−1^, and 5.8 × 10^−39^ C∙m^2^∙V^−1^, respectively. It can be found that their determinant of permittivity is the dipole polarizability of these polyimides. Therefore, as the dipole polarizability of C0-SPI, C1-SPI, and C2-SPI enhances in turn, their dielectric constants also increase. 

Since the dipole polarization is directly related to the dipole moment of the electric field, not only the value but also the angle induced by the electric field of the dipole moment affect the polarizability. The induced angle is determined by the motility of the polymer chains, so it is further studied based on the simulation results. The parameter root-mean-square fluctuation (RMSF) of the component was used to characterize the motion range of the polymers. The RMSF of the main chains of C0-SPI, C1-SPI, and C2-SPI was 0.0682 nm, 0.0608 nm, and 0.0585 nm, while the RMSF of the side chains was 0.115 nm, 0.108 nm, and 0.111 nm, respectively. It can be found that the motility of the main chains is very low, which is consistent with the glassy state of the polymers. The motility of the side chains is much higher than that of the main chains, indicating that the rotation of the side groups decides the orientation of the dipoles in the three polyimides. Because of the similar polar groups in the side chains, their orientation ability is the determinant of the dipole polarization. Each group on the side chains was decomposed to calculate the RMSF. Among them, -CH_2_-(1) is the methylene directly connected to the benzene ring, while -CH_2_-(2) is the methylene connected to -CH_2_-(1). With the enhancement of the flexibility of the side chains, the RMSF of the corresponding groups in C0-SPI, C1-SPI, and C2-SPI increases (Appendix A), indicating the larger range for the groups. It can also be confirmed through the stacking illustration of the samples (Appendix A). Because the direction of the applied electric field is unique and random during polarization, the larger motion range of the dipole is beneficial for their orientation along the electric field, resulting in a stronger polarization.

The polarization ability of dielectrics can also be characterized by their dipole moment variation under an external electric field. The variations in all of the conformations in Appendix A in three mutually perpendicular directions were calculated by performing quantum chemical simulations, and the average variations for each conformation, Δ*μ*_x,_ Δ*μ*_y_, and Δ*μ*_z,_ were obtained by weighing the proportion of them. Δ*μ*_avg_ is the arithmetic mean of Δ*μ*_x,_ Δ*μ*_y_, and Δ*μ*_z,_ revealing the dipole moment variations of the molecules completely. The results are summarized in Appendix A, in which C2-SPI with the most flexible side groups showed the highest Δ*μ*_avg_. Thus, C2-SPI exhibited the strongest polarization ability, consistent with the highest permittivity.

### 2.3. Energy Storage Capacity of Polyimides

The high-field dielectric properties of C0-SPI, C1-SPI, and C2-SPI at room temperature were studied through bipolar *D*–*E* loop measurements (Figure 5a,c,e). C0-SPI, C1-SPI, and C2-SPI all showed narrow bipolar *D*–*E* hysteresis loops at a low electric field, suggesting that no obvious ferroelectric domains form in the three high dielectric materials. The maximum tolerated electric fields of C0-SPI, C1-SPI, and C2-SPI are 400 MV∙m^−1^, 500 MV∙m^−1^, and 550 MV∙m^−1^, respectively. At a high field, the materials show broader and asymmetric loops with stronger remnant polarization, because the dipoles with a longer relaxation time are oriented. The discharge energy density *U*_e_ and charge–discharge efficiency *η* of the materials were obtained from the *D*–*E* hysteresis loops (Figure 5b,d,f). The maximum discharge energy densities *U*_e_ of C0-SPI and C1-SPI are 5.75 J∙cm^−3^ and 8.16 J∙cm^−3^. For C2-SPI, because it has the highest permittivity and breakdown strength, the *U*_e_ reached 10.0 J∙cm^−3^. The charge–discharge efficiency *η* of C0-SPI, C1-SPI, and C2-SPI is 79%, 80%, and 70%, respectively. The efficiency *η* slightly reduces when increasing the electrical field because of the loss from the orientation polarization and electronic conduction. Compared with inorganic dielectric oxide, the polyimides achieve both a high breakdown electric field and self-standing property. Furthermore, the good flexibility and mechanical properties of the polymers are beneficial to the winding process of the capacitors, realizing the miniaturization of electronic devices [59,60].

As the three high dielectric polyimides all show outstanding thermal stability, the energy storage properties of C0-SPI, C1-SPI, and C2-SPI at various temperatures are further discussed, as shown in Figure 6. Even at 200 °C, the electrical breakdown of the three materials can reach more than 300 MV∙m^−1^, among which the breakdown voltage of C1-SPI reaches more than 350 MV∙m^−1^. At the highest electric field, the *D*–*E* hysteresis loops of each material at various temperatures basically coincide, indicating that the charge–discharge properties of the three materials remain stable even at high temperatures. The maximum discharge energy density *U*_e_ of C0-SPI, C1-SPI, and C2-SPI is 3.04 J∙cm^−3^, 3.68 J∙cm^−3^, and 2.59 J∙cm^−3^, respectively. In addition, their charge–discharge efficiency *η* is 80%, 84%, and 71%. Based on the result above, three novel high-*k* polyimides are promising candidates for high-temperature capacitors. 

## 3. Conclusions

In summary, three new polyimides C0-SPI, C1-SPI, and C2-SPI with the same rigid backbone and different linking groups to the dipoles were successfully synthesized via monomer design and polycondensation. The polyimides all show amorphous states and can be prepared as self-standing films with a large area, which showed good film forming properties and a high breakdown field. With the rigid backbones, they show excellent thermal properties (*T*_d5%_ > 400 °C and *T*_g_ > 300 °C) and good dimensional stability. The different linking groups endow the dipole side groups with different flexibility, showing that with the increase in the flexibility the dielectric constant increases successively, where the dielectric constant of C0-SPI, C1-SPI, and C2-SPI is 5.6–5.2, 6.0–5.3, and 6.5–6.0 from 100 Hz to 1 MHz. Further experimental and theoretical studies on the dielectric properties of the three materials show that the enhancement of the flexibility of the polar side chains endows the dipole with a larger range of motion, leading to a stronger orientation ability under the external electric field, which is beneficial to the effective polarization of the materials. The three materials were applied to the capacitor energy storage device at room temperature and all show high discharge energy density *U*_e_ of 5.75 J∙cm^−3^, 8.16 J∙cm^−3^, and 10.0 J∙cm^−3^, respectively. Due to their excellent thermal stability, the three materials can work normally even at 200 °C, delivering *U*_e_ of 3.04 J∙cm^−3^, 3.68 J∙cm^−3^ and 2.59 J∙cm^−3^, respectively. This work achieved the simple regulation and control of dielectric properties of materials and provides a possible synthesis strategy for designing high-*k* polymers.

## 4. Materials and Methods

### 4.1. Materials

Sodium thiomethoxide and N, N-dimethylformamide (DMF) were purchased from Energy Chemical (Shanghai, China) and used as received. 1,3-Dibromo-5-fluorobenzene, tetrakis(triphenylphosphine)-palladium (Pd(PPh_3_)_4_), N-bromosuccinimide (NBS), 4-aminophenethyl alcohol, and 3,5-dibromotoluene were purchased from Bide Pharmatech Ltd. (Shanghai, China) and used as received. Pyromellitic dianhydride (PMDA) was purchased from TCI Shanghai (Shanghai, China) and used as received. Sodium methansulfinate was purchased from Accela ChemBio Co., Ltd. (Shanghai, China) and used as received. 4-Aminophenylboronic acid hydrochloride was purchased from Hebei Summedchem Co., Ltd. (Xingtai, China) and used as received. Ethyl acetate (EA), petroleum ether (PE), and dichloromethane (DCM) were purchased from Shanghai Titan Scientific Co., Ltd. (Shanghai, China) and used as received. Aliquat 336, triphenylphosphine, 2,2′-azobis(2-methylpropionitrile) (AIBN), and trifluoroacetic acid were purchased from the Shanghai Aladdin Biochemical Technology Co., Ltd. (Shanghai, China) and used as received. All other reagents and solvents were obtained from the Guangzhou Chemical Reagent Factory (Guangzhou, China) without further purification.

### 4.2. Synthesis of Monomers and Polyimides

The synthetic route and characterization of the three diamine monomers (C0-STPDA, C1-STPDA, and C2-STPDA) are given in Appendix A. The chemical structures of the three diamines were confirmed through ^1^H NMR spectra, ^13^C NMR spectra, mass spectra, and elemental analysis. Polycondensation with PMDA obtained three polyimides C0-SPI, C1-SPI, and C2-SPI. The polyimide films were prepared via a traditional two-step procedure. As a typical example, the polymerization of diamine C1-STPDA and PMDA was carried out according to the following procedure: C1-STPDA (0.5 g, 1.42 mmol) was dissolved in 4 mL anhydrous DMF, then PMDA (0.309 g, 1.42 mmol) was added in. The mixture was stirred at room temperature overnight to form a viscous poly (amic acid) (PAA) solution. The PAA solution was coated uniformly onto a clean and dry glass, and thermally imidized in a vacuum oven with the temperature program of 80 °C (1 h)/150 °C (1 h)/250 °C (1 h)/300 °C (1 h) to produce the C1-SPI film. Between two temperature steps, the heating rate in this work is 5 °C∙min^−1^. The film thickness is controlled between 10 and 15 μm by the coating machine. All the three polyimides can be prepared into large-area films.

### 4.3. Instrumentation

All NMR spectra of the intermediates and monomers were recorded on a Bruker Model AVANCE III 400 spectrometer. Samples (5 mg of each compound) were dissolved in 0.5 mL of deuterated dimethyl sulfoxide (DMSO-*d*_6_) or deuterated chloroform (CDCl_3_) using tetramethylsilane (TMS) as the internal reference. Mass spectra were measured on a Thermo EI mass spectrometer (DSQ II). Elemental analysis of monomers and polyimides were both performed on a CHN elemental analyzer. Fourier-transform infrared (FT-IR) spectra were recorded on a Bruker Tensor 27 spectrometer using attenuation total reflection (ATR). Wide-angle X-ray diffraction (WAXD) was measured on a Rigaku SmartLab X-ray diffractometer. The range of the scan angle was 10°–60°, and the scan speed was 10°∙min^−1^. Thermogravimetric analyses (TGA) were performed with a PerkinElmer PE Pyris1 TGA with a heating rate of 20 °C∙min^−1^ from 50 °C to 800 °C under nitrogen (3–5 mg of each sample). Differential scanning calorimetry (DSC) curves were obtained with a PerkinElmer DSC-8500 thermal analyzer at a heating rate of 10 °C min^−1^ from 50 °C to 300 °C under nitrogen. Gel permeation chromatography (GPC) was measured via a Waters Model 717 plus autosampler with a Waters Model 1515 isocratic HPLC pump. DMF was used as the eluent, and relative molecular weight was calibrated with narrow polydispersity polystyrene standards. Thermal mechanical analysis (TMA) was conducted through a TMA Q400 analyzer at a preload force of 0.05 N and a heating rate of 10 °C∙min^−1^ from 50 °C to 350 °C. 

Broadband dielectric spectroscopy (BDS) measurements were conducted on a Solartron SI 1260 impedance/gain phase analyzer from 10^2^ to 10^6^ Hz at room temperature and on a Triton DS-6000 dielectric thermal performance spectrometer at various temperatures ranging from −100 °C to 200 °C. The applied voltage was 2 V_rms_ (root-mean-square AC voltage). Gold (1 cm diameter) was deposited onto both surfaces of the films as electrodes using an SBC-12 ion sputter coater. Electric displacement–electric field (*D*–*E*) hysteresis loop measurements were carried out using a Premiere II ferroelectric tester from Radiant Technologies in combination with a Trek 10/10B-HS high voltage amplifier. The tests were performed at 100 Hz, and the films were immersed in silicone oil to avoid corona discharge in the air. The temperature was controlled via MR Hei-Tec magnetic stirrers from Heidolph. 

## Figures and Tables

**Figure 1 molecules-27-06337-f001:**
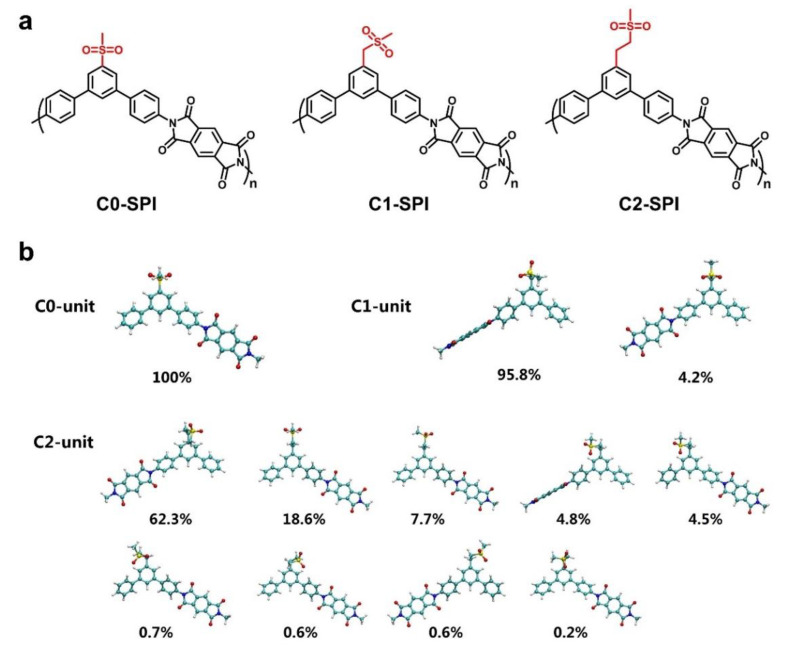
(**a**) Chemical structures of the three polyimides C0-SPI, C1-SPI, and C2-SPI. (**b**) Various conformations of C0 unit, C1 unit, and C2 unit. According to the Boltzmann distribution, the proportion of each conformation was calculated using their Gibbs free energy.

**Figure 2 molecules-27-06337-f002:**
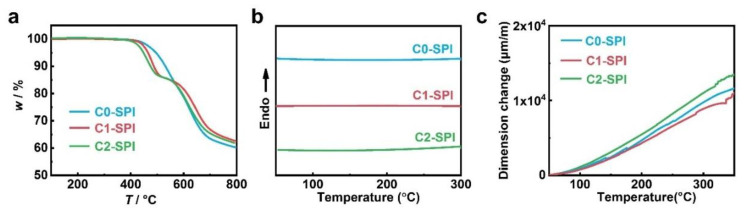
(**a**) TGA curves, (**b**) DSC curves, and (**c**) TMA curves of the three polyimides C0-SPI, C1-SPI, and C2-SPI.

**Figure 3 molecules-27-06337-f003:**
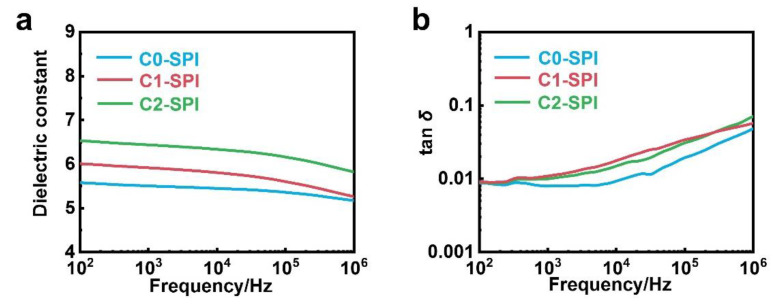
(**a**) Dielectric constant and (**b**) dielectric loss as a function of frequency at room temperature of C0-SPI, C1-SPI, and C2-SPI.

**Figure 4 molecules-27-06337-f004:**
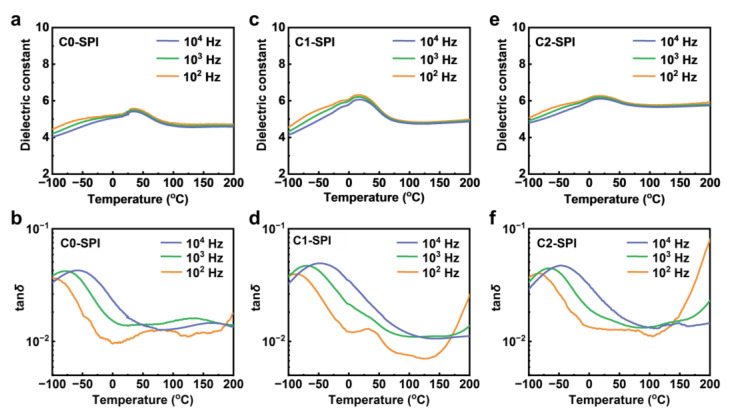
(**a**,**c**,**e**) Dielectric constant and (**b**,**d**,**f**) dielectric loss versus temperature at various frequencies of C0-SPI, C1-SPI, and C2-SPI, respectively.

**Figure 5 molecules-27-06337-f005:**
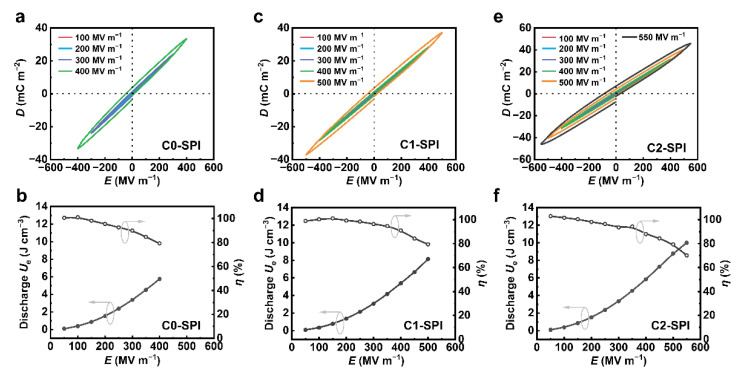
Bipolar *D*–*E* loops for (**a**) C0-SPI, (**c**) C1-SPI, and (**e**) C2-SPI, and corresponding discharged energy density (*U*_e_) and charge–discharge efficiencies (*η*) as a function of the poling field (**b**,**d**,**f**).

**Figure 6 molecules-27-06337-f006:**
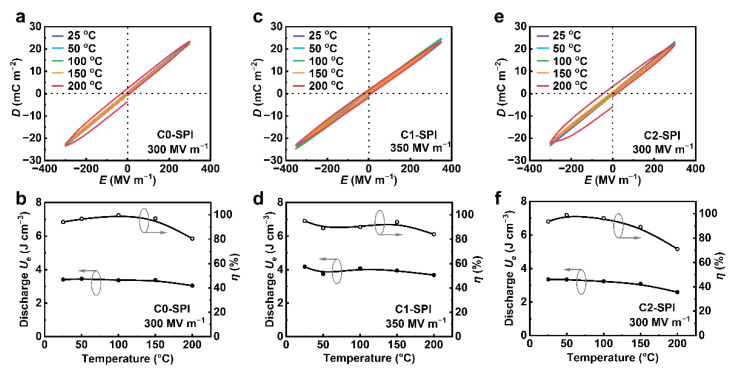
Bipolar *D*–*E* loops at different temperatures of (**a**) C0-SPI at 300 MV m^−1^, (**c**) C1-SPI at 350 MV∙m^−1^, and (**e**) C2-SPI at 300 MV∙m^−1^, and corresponding discharged energy density (*U*_e_) and charge–discharge efficiencies (*η*) as a function of temperature (**b**,**d**,**f**).

## Data Availability

The data presented in this study are available upon request from the corresponding author.

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
