# Peer review of "Temperature-Resistant Intrinsic High Dielectric Constant Polyimides: More Flexibility of the Dipoles, Larger Permittivity of the Materials"

_molecules, 2022, doi:10.3390/molecules27196337_

Round 1
Reviewer 1 Report
In this work, the authors designed three kinds of PIs with same rigid backbone and different linking groups to the dipoles and compared their dielectric properties and energy storage performance. The research is meaningful and the manuscript is well organized. Nevertheless, the authors should address the following comments before a final acceptance.
1. What is the test frequency of the D-E hysteresis loop? 10 Hz or 100 Hz. It is better to add this information in the Experimental section.
2. It is observed that all the bipolar D-E hysteresis loops in Figures 5 and 6 are not symmetric. The author should briefly explain the phenomena. Regarding the discharged Ue and charge-discharge efficicency, which parts of the loops are used to do the calculations?
Author Response
In this work, the authors designed three kinds of PIs with same rigid backbone and different linking groups to the dipoles and compared their dielectric properties and energy storage performance. The research is meaningful and the manuscript is well organized. Nevertheless, the authors should address the following comments before a final acceptance.
1. What is the test frequency of the D-E hysteresis loop? 10 Hz or 100 Hz. It is better to add this information in the Experimental section.
A: Thank you so much for your careful check. The test frequency of the D-E loop is 100 Hz in this study. The corresponding information has been added in the revised manuscript, as shown in L334, P9.
2. It is observed that all the bipolar D-E hysteresis loops in Figures 5 and 6 are not symmetric. The author should briefly explain the phenomena. Regarding the discharged Ue and charge-discharge efficiency, which parts of the loops are used to do the calculations?
A: Thank you for the good questions! Due to the dielectric loss contributing by orientation polarization and electronic conduction, the electric displacement and the electric field strength will deviate from the linear relationship. Especially at high field, the materials will show strong remnant polarization because the dipoles with a longer relaxation time are oriented, leading to the asymmetric loops. It also can be confirmed that the bipolar D-E hysteresis loops in Figures 5 and 6 are symmetric at low electric field strength, while asymmetric at high electric field strength. The discharged Ue and charge-discharge efficiency of the polymers are calculated by the positive electric field strength part of the loops, corresponding to a single charge and discharge process starting from zero electric field. This calculation method is also the same as some literatures.[1] The corresponding information has been added in the revised manuscript, as shown in L219 and L223, P6.
[1] Zhu, L.; Wang, Q., Novel Ferroelectric Polymers for High Energy Density and Low Loss Dielectrics. Macromolecules 2012, 45, 2937-2954.
Reviewer 2 Report
The authors synthesized and investigated three polyimides 14 (C0-SPI, C1-SPI, and C2-SPI), characterized their structures using various techniques and systematically measured their dielectric properties. In particular, with the enhancement of dielectric constant, the polyimides are found to exhibit high energy storage density. The paper is very well written and is highly recommended for publication.
However, as one additional supplement, I would recommend the authors to compare their energy-storage performance with that of the dielectric oxides such as SrTiO3 and Na0.5BiTiO3 in the discussion paragraph. For example, the energy-storage density, breakdown field, fatigue performance, etc. The relevant information was summarized and reviewed here, e.g., Materials 14, 7854 (2021); Adv. Energy Mater. 12, 2201199 (2022). Such comparative discussion can help the readers to better know the pros and cons of different material systems in electrostatic energy storage.
Author Response
The authors synthesized and investigated three polyimides 14 (C0-SPI, C1-SPI, and C2-SPI), characterized their structures using various techniques and systematically measured their dielectric properties. In particular, with the enhancement of dielectric constant, the polyimides are found to exhibit high energy storage density. The paper is very well written and is highly recommended for publication.
However, as one additional supplement, I would recommend the authors to compare their energy-storage performance with that of the dielectric oxides such as SrTiO3 and Na0.5BiTiO3 in the discussion paragraph. For example, the energy-storage density, breakdown field, fatigue performance, etc. The relevant information was summarized and reviewed here, e.g., Materials 14, 7854 (2021); Adv. Energy Mater. 12, 2201199 (2022). Such comparative discussion can help the readers to better know the pros and cons of different material systems in electrostatic energy storage.
A: Thank you for your constructive suggestion. According to the literatures, the dielectric oxide bulks show low breakdown electric field. Most of them are lower than 100 MV m-1. The nanoscale ceramic films can withstand stronger electric fields (> 200 MV m-1), but they are not self-standing. By contrast, the polyimides in this article achieve both high breakdown electric field (> 400 MV m-1) and self-standing property. Furthermore, the good flexibility and mechanical properties of the polymers is beneficial to the winding process of the capacitors, realizing the miniaturization of electronic devices. So, we believe that the polymer dielectrics are more promising materials for energy storage. The corresponding information has been added in the revised manuscript, as shown in L232, P6
[1] Wei, X. K.; Dunin-Borkowski, R. E.; Mayer, J., Structural Phase Transition and In-Situ Energy Storage Pathway in Nonpolar Materials: A Review. Materials (Basel) 2021, 14, 7854.
[2] Wei, X. K.; Domingo, N.; Sun, Y.; Balke, N.; Dunin‐Borkowski, R. E.; Mayer, J., Progress on Emerging Ferroelectric Materials for Energy Harvesting, Storage and Conversion. Adv. Energy Mater. 2022, 12, (24), 2201199.